# Granzyme K initiates IL-6 and IL-8 release from epithelial cells by activating protease-activated receptor 2

Dion Kaiserman[1]*, Peishen Zhao[2,3], Caitlin Lorraine Rowe[1], Andrea Leong[2,3], Nicholas Barlow[2], Lars Thomas Joeckel[1], Corinne Hitchen[1], Sarah Elizabeth Stewart[1¤], Morley D. Hollenberg[4], Nigel Bunnett[5], Andreas Suhrbier[6,7], Phillip Ian Bird[1]

1 Department of Biochemistry & Molecular Biology, Biomedicine Discovery Institute, Monash University, Clayton, VIC, Australia, 2 Monash Institute of Pharmaceutical Sciences, Monash University, Parkville, VIC, Australia, 3 Australian Research Council Centre of Excellence in Convergent Bio-Nano Science and Technology, Monash University, Parkville, VIC, Australia, 4 Department of Physiology & Pharmacology, Department of Medicine, University of Calgary, Calgary AB, Canada, 5 Department of Molecular Pathobiology, Department of Neuroscience and Physiology, Neuroscience Institute, New York University, New York, NY, United States of America, 6 QIMR Berghofer Medical Research Institute, Brisbane, QLD, Australia, 7 Australian Infectious Disease Research Centre, University of Queensland, Brisbane, QLD, Australia

¤ Current address: Department of Biochemistry and Genetics, La Trobe Institute for Molecular Science, La Trobe University, Melbourne VIC, Australia
* dion.kaiserman@monash.edu

**Data Availability Statement:** All relevant data are within the paper and its Supporting Information files.

## Abstract

Granzyme K (GzmK) is a tryptic member of the granzyme family of chymotrypsin-like serine proteases produced by cells of the immune system. Previous studies have indicated that GzmK activates protease-activated receptor 1 (PAR1) enhancing activation of monocytes and wound healing in endothelial cells. Here, we show using peptides and full length proteins that GzmK and, to a lesser extent the related protease GzmA, are capable of activating PAR1 and PAR2. These cleavage events occur at the canonical arginine P1 residue and involve exosite interactions between protease and receptor. Despite cleaving PAR2 at the same point as trypsin, GzmK does not induce a classical $Ca^{2+}$ flux but instead activates a distinct signalling cascade, involving recruitment of β-arrestin and phosphorylation of ERK. In epithelial A549 cells, PAR2 activation by GzmK results in the release of inflammatory cytokines IL-6 and IL-8. These data suggest that during an immune response GzmK acts as a pro-inflammatory regulator, rather than as a cytotoxin.

## Introduction

Granzymes (Gzm) comprise a family of chymotrypsin-like serine proteases primarily produced by cells of the immune system. There are five human granzymes, each with a distinct substrate specificity: GzmA (dimeric tryptase), GzmB (asp-ase), GzmH (chymase), GzmK (monomeric tryptase), and GzmM (met-ase). Classically, granzymes are described as intracellular cytotoxins that kill compromised cells by activating redundant death pathways when delivered by the pore-forming protein, perforin [1]. More recently, however, numerous non-

**Funding:** This work was funded by a National Health and Medical Research Council Australia Project Grant 1141421 (PIB & AS), a National Health and Medical Research Council Australia Program Grant 490900 (PIB), National Institutes of Health grants NS102722, DE026806, DK118971 and DE029951 (NB) and Department of Defense grant W81XWH1810431 (NB). The funders had no role in study design, data collection and analysis, decision to publish, or preparation of the manuscript.

**Competing interests:** The authors have declared that no competing interests exist.

cytotoxic and extracellular roles have been reported for granzymes including: cleavage of extracellular matrix to facilitate migration, direct antiviral effects, immune signalling via activation of cytokines, and activation of cell surface receptors [2].

The protease-activated receptors (PARs) are a family of four G-protein coupled receptors that respond to protease activity. Proteolytic cleavage at a specific site within the amino (N) terminal extracellular domain of the receptor generates a new N-terminus, exposing a short peptide that acts as a tethered ligand to activate the receptor and initiate downstream signalling cascades. PAR activation typically initiates intracellular $Ca^{2+}$ mobilisation and pERK activation, as well as other regulatory pathways including arrestin-recruitment, receptor internalisation and degradation.

Canonical activating cleavage sites of all PARs have been mapped and contain a basic residue at the P1 site. PAR1 is activated by thrombin at Arg41, PAR2 is activated by trypsin at Arg36, and both thrombin and trypsin activate PAR4 at Arg47. PAR3 is also activated by thrombin, at Lys38, but does not appear to signal [3]. As evident from the example of PAR2, distinct proteases may cleave at the same P1 residue within the extracellular domain of a PAR with the same outcome, or they may cleave at distinct residues, resulting in alternative tethered ligand sequences that can activate unique signalling networks. In addition, other cleavage sites, distant from the canonical P1 activation site, may also act to disarm the receptor and prevent receptor activation [3].

There are a number of reports of PAR activation by granzymes. GzmA activates the thrombin receptor (PAR1) on neural cells [4], and cleaves a different receptor, likely PAR2, on monocytes to release cytokines [5]. GzmB cleaves PAR1 and PAR2 to mediate biological effects, however these sites have not been mapped [6,7]. GzmK cleaves and activates PAR1 on endothelial cells and fibroblasts [8–10]. Here we show that GzmK is also capable of cleaving and activating PAR2 on the surface of epithelial cells, invoking cytokine release. Although the cleavage occurs at the canonical site, $Ca^{2+}$ mobilisation is not initiated. Instead, signalling is biased toward ERK phosphorylation and β-arrestin recruitment.

## Materials and methods

### Cells and cell culture

A549 and COS-1 cells were grown as adherent monolayers in DMEM supplemented with 2 mM L-glutamine, 50 units/ml penicillin and 50 μg/ml streptomycin. HEK293 and KNRK cells were grown in DMEM supplemented with 10% FBS. Cells were passaged at ~80% confluency by washing monolayers with PBS and detaching from the growth support with 0.25% (w/v) trypsin and replating into fresh medium.

### Recombinant proteins

Human GzmA and GzmB were purified as previously described [11]. The human GzmK cDNA was modified to replace the signal peptide and prodomain with a 6 × His tag and bovine enterokinase cleavage site and cloned into the *Escherichia coli* expression plasmid pEXP-His. Recombinant protein was expressed in inclusion bodies in BL21 arabinose-inducible *E. coli*. These were collected by centrifugation of the lysed bacterial pellet and washed three times: wash 1 contained 50 mM Tris, pH 7.4, 20 mM EDTA, 20 mM DTT, 1% (v/v) Triton X-100; wash 2 contained 50 mM Tris, pH 7.4, 20 mM EDTA, 1 M NaCl; and wash 3 contained 50 mM Tris, pH 7.4, 20 mM EDTA. The insoluble protein pellet was collected and denatured in 7 M guanidine, 100 mM DTT, and 100 mM Tris, pH 8.3.

The denatured protein was refolded by slow, drop-wise addition into 600 mM L-arginine, 50 mM Tris, pH 7.4, 500 mM NaCl, 1 mM EDTA, 10% (v/v) glycerol, 1 mM L-cysteine and

incubated at room temperature for 72 h. The solution containing soluble and correctly folded pro-granzyme was concentrated by tangential flow filtration before activation as described for GzmA and GzmB.

Granzymes were assayed for activity by gel shift assay with serpin inhibitors and were at least 95% active. Where indicated, granzymes were inactivated by reaction with a 100-fold molar excess of the irreversible serine protease inhibitor, aminoethyl benzene sulfonyl fluoride (AEBSF), for 3 h at 37˚C.

For cytokine release assays, endotoxin was removed by adding Triton X-114 to a final concentration of 1% (v/v), incubating at 4˚C for 30 min, then 37˚C for 10 min followed by centrifugation at 16,000 $g$ for 10 min at RT and retaining the upper, aqueous phase. This was repeated 5 times and the resultant preparation assayed for endotoxin using the Pyrogene™ Recombinant Factor C Endotoxin Detection Assay (Lonza Bioscience). All proteases contained <0.05 EU/ml endotoxin.

## Peptide cleavage assays

Peptides corresponding to N-terminal fragments of human PAR1 and PAR2 (350 μM) were incubated with granzymes (100 nM) in Hank's Balanced Salt Solution (HBSS) pH 7.4 for 0, 2, 4, and 15 hours at 37˚C. Reactions were quenched with an equal volume of 50% acetonitrile and 0.1% trifluoroacetic acid in water. Degradation was assessed using an Agilent 1260 Infinity HPLC System with Poroshell 120, SB-C18, 2.1 × 30 mm, 2.7 μM column, 5–95% acetonitrile in water over 9 min, 0.1% TFA throughout. The reaction products were identified by mass spectrometry using a Shimadzu LCMS 2020, single quadrupole in electrospray in positive ionization mode with a mass range of 200–2000 $m/z$.

## Construction and cleavage of NanoLuc®-PAR constructs

The pcDNA3/nLuc®-PAR1-eYFP and pcDNA3/nLuc®-PAR2-eYFP vectors were a gift from Prof. M. Hollenberg (University of Calgary). The consensus site mutations were generated by Q5 site-directed mutagenesis (New England Biolabs) using primers 5'-CTTAGATCCCGCGT-CATTTCTTCTCAG for PAR1 or 5'- CTTAGATCCCGCTAGCTTTCTTCTCAGGAACCCCAATG and 5'-GTGGCATTTGTTGCTTTG for PAR2. COS-1 cells were transfected in 24-well plates by adding 1–2 μg of each plasmid to $5 \times 10^5$ COS-1 cells in DMEM supplemented with 2 mM l-glutamine, 50 units/ml penicillin, 50 μg/ml streptomycin, 400 μg/ml DEAE-dextran, and 25 μM chloroquine for 3 h at 37˚C. The medium was replaced with 10% dimethyl sulfoxide in DMEM for 2 min, and cells then returned to complete DMEM.

48 h later cells were washed with PBS and incubated at 37˚C with a serial dilution of granzyme in 500 μl serum free DMEM. Supernatants were collected after 2 h and centrifuged for 5 min at 16,000 g to remove cellular debris. 80 μl was transferred to an opaque 96-well tray and assayed for luciferase activity. Nano-Glo® Luciferase Assay Substrate was diluted in 50 volumes of Nano-Glo® Luciferase Assay Buffer and 20 μl of dilute substrate was mixed with clarified supernatant for 5 min and luminescence measured on a FLUOstar OPTIMA microplate reader (BMG Labtech). Experiments were repeated 3–5 times and are presented as mean ± standard deviation.

## Calcium mobilisation assays

KNRK cells and KNRK cells stably expressing human PAR2 [12] were seeded into poly-lysine coated 96 well plates ($3 \times 10^5$ cells/well). After 24 h, cells were loaded with of the calcium sensitive dye FURA2/AM (1 μM) [Invitrogen, F-1221] in $Ca^{2+}$ buffer (150 mM NaCl, 2.6 mM KCl, 0.1 mM $CaCl_2$, 1.18 mM $MgCl_2$, 10 mM D-glucose, 10 mM HEPES, pH 7.4) containing 4 mM

Probenecid [Sigma Aldrich, P8761] for 1 h at 37˚C. Fluorescence was measured at 340 nm and 380 nm excitation and 530 nm emission using a FlexStation Microplate Reader (Molecular Devices). After a baseline reading for 60 s, cells were treated with vehicle, GzmK (100 nM) or inactive GzmK (100 nM), while trypsin (10 nM) was the positive control, and reading continued for up to 5 min.

To examine whether GzmK has inhibitory effects on trypsin-mediated $Ca^{2+}$ signalling, cells were pre-treated with GzmK or its inactive form (100 nM) for 30 min, washed three times with assay buffer and challenged with trypsin (10 nM). Areas under the curve of protease-mediated $Ca^{2+}$ responses were normalized as the percentage of trypsin response.

To measure endogenous PAR1 and PAR2 activation in A549 cells, cells were seeded into 96 well plates at a density of $3 \times 10^4$ cells/well 24 h before the measurement. Cells were then incubated with Fluo-8AM (10 µM) for 1 h at 37˚C. Plates were read using the FDSS/µCELL (Hamamatsu, Japan), with 480 nm excitation 542 nm emission filters every 10 s for 2 min including a 15 s baseline. PAR1 and PAR2 activating peptides were added at 15 s at indicated concentrations. 10 µM ATP was used as a positive control.

### ERK phosphorylation assays

KNRK and KNRK-PAR2 cells were seeded into 96 well plates ($3 \times 10^5$ cells/well). After 24 h, cells were serum starved in 80 µl phenol red free DMEM for at least 12 h to reduce basal ERK activity. Cells were challenged with 10 µl of buffer, GzmK (100 nM), with or without 30 min pre-treatment of EGFR inhibitor AG1478 (100 nM), inactivated GzmK (100 nM), trypsin (10 nM), FBS (10%) for 5 min. Concentration response measurements were also conducted for trypsin and GzmK at increasing protease concentrations. ERK1/2 phosphorylation was detected using the AlphaScreen SureFire ERK1/2 assay kit (PerkinElmer) according to the manufacturer's instructions.

### β-arrestin BRET assays

BRET assays were performed essentially as described [13]. Briefly, HEK 293 cells were plated into 10 cm dishes at a density of $2 \times 10^6$ cells/plate and transfected with 1 µg PAR2-Rluc8 and 4 µg β-arrestin$_1$-YFP using polyethylenimine (PEI) at a ratio of 1:6 (DNA:PEI). 24 h post-transfection, cells were plated onto a white 96 well plate at a density of $3 \times 10^4$ cells/well. 24 h post plating, the medium was replaced with 80 µl 1 × Hank's Buffered Salt Solution, pH 7.4, at 37˚C for 30 min, and treated with either buffer, trypsin (10 nM), GzmA (100 nM) or GzmK, (100 nM) for 2 h. 5 µM coelenterazine was then added and incubated for 5 min. Bioluminescence was measured at 475 ± 30 nm (Rluc8) and fluorescence at 530 ± 30 nm (YFP) on a LUMIstar Omega (BMG Labtech). β-arrestin$_1$ recruitment is indicated as the ratio change between the two wavelengths.

### Cytokine ELISAs

A549 cells were seeded in 12 well plates at a density of $0.5 \times 10^6$ cells/ml in 1 ml DMEM and incubated at 37˚C for 6 h to allow cell adhesion. DMEM was then exchanged for 1 ml SF-DMEM and cells incubated overnight at 37˚C. The next day, medium was replaced with 500 µl of SF-DMEM containing Gzm or PAR activating peptide for 24 h at 37˚C. Medium was collected and cellular debris removed by centrifugation at 16,000 *g* for 5 min.

ELISA was performed using the human IL-6, human IL-8, human TNFα, and human IL-1β sandwich ELISAs (Life Technologies) according to the manufacturer's instructions. ELISA plates were coated overnight with 100 µl capture antibody in PBS. Plates were washed 3 times with PBS + 0.05% (v/v) Tween-20, then incubated with 200 µl sample diluted in PBS for 2 h at

room temperature. Wells were washed 5 times with PBS, then incubated with 100 μl detection antibody in PBS for 1 h at room temperature. Antibody was removed and wells washed 5 times with PBS + 0.05% Tween-20 and then incubated with streptavidin conjugated to horse-radish peroxidase at room temperature for 30 min. Cells were washed 5 times with PBS + 0.05% Tween-20 and then incubated with 100 μl 3,3',5,5'-tetramethylbenzidine and colour development monitored over 15 min. Reactions were stopped by addition of 50 μl $H_2SO_4$ and colour measured on a FluoStar Omega plate reader at both 450 nm (measurement) and 570 nm (background). Background corrected optical density for untreated cells was subtracted from each measurement and concentration of cytokine was then interpolated from a standard curve. Cytokine release was plotted against treatment concentration and linear regression applied to derive cytokine release per concentration of treatment. ELISA experiments were performed 2–4 times and are presented as mean ± standard deviation.

## Results

### Cleavage of full-length PAR1 and PAR2 by GzmA and GzmK

Previous studies have indicated that GzmA and GzmK can cleave and activate PARs [4,9,10,14]. To replicate and extend these findings we evaluated GzmA- and K-mediated cleavage of the best-described PARs, PAR1 and PAR2. We examined transiently expressed Nano-Luc®-PAR-eYFP constructs in COS-1 cells. In this system luciferase is fused to the N-terminus of the PAR, and yellow fluorescent protein (YFP) is fused to the C-terminus to report receptor localization. Treatment of transfected cells with an exogenous protease that cleaves the PAR extracellular N-terminal domain liberates luciferase, which can then be measured in the medium. To ensure that any cleavage detected occurs at the canonical activation site, we also generated constructs where the Arg41 residue in PAR1, or Arg36 residue in PAR2, was mutated to Ala. We also included enzymatically inactive forms of the granzymes (GzmA-AI and GzmK-AI) in the experiment to confirm that any effects required proteolytic activity as opposed to protein-protein interactions.

We first tested the ability of granzymes to liberate luciferase from NanoLuc®-PAR1-eYFP (Fig 1A). Both GzmA and GzmK cleaved PAR1 in this system. Cleavage by GzmK was more efficient than GzmA, releasing 6 to 7-fold more luciferase (Fig 1A). Inactivation of GzmA or GzmK blocked their ability to release the NanoLuc®, confirming that this is an enzymatic reaction (Fig 1A).

To determine whether cleavage occurred at the consensus site (Arg41) we mutated this residue to alanine (Arg41Ala). As a control for protease cleavage away from the consensus site, we used GzmB, which has been shown to cleave PAR1 [6], but its requirement for an acidic P1 residue suggests it does not cleave at the consensus Arg41. As shown in Fig 1A, GzmB released luciferase from cells, and release was not prevented when PAR1 carried the Arg41Ala (Fig 1A).

Alanine substitution of the P1 residue (Arg41Ala) abolished cleavage by GzmK, confirming that this is the true cleavage point. By contrast, GzmA cleavage was not affected by the Arg41Ala mutation (Fig 1A). This indicates that either GzmA does not cleave at the canonical site, or exosite interactions are sufficient to overcome the decreased affinity caused by loss of Arg41.

In cells expressing NanoLuc®-PAR2-eYFP, both GzmA and GzmK cleaved the full-length protein, and we once again noted that GzmK was significantly more efficient, releasing 4-fold more luciferase (Fig 1B). Importantly, GzmK cleavage was completely abolished by the Arg36Ala P1 mutation. By contrast with PAR1, GzmA cleavage was significantly reduced by the P1 mutation, indicating that GzmA does cleave at the consensus site in PAR2 (Fig 1B).

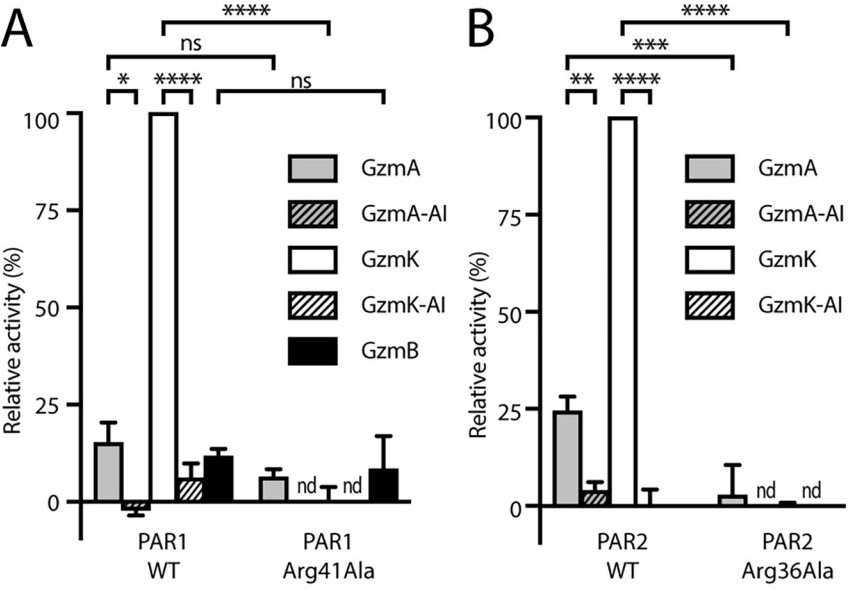

**Fig 1. Cleavage of full-length PARs by granzymes.** COS-1 cells were transfected with (A) NanoLuc®-PAR1-eYFP; or (B) NanoLuc®-PAR2-eYFP plasmids. Cells were treated with a dilution series of the indicated granzyme or AEBSF-inhibited granzyme in serum-free DMEM for 2 h at 37°C. Supernatants were assayed for NanoLuc® activity. Data are expressed as a percentage of GzmK-treated WT PAR1 or PAR2. Pairwise Student's t-tests were used to determine significant differences. nd, not determined; ****, $p < 0.0001$; ***, $p < 0.001$; **, $p < 0.01$; *, $p < 0.05$; ns, $p > 0.05$. The

extracellular N-terminal sequences of (C) PAR1 or (D) PAR2 were represented by sets of overlapping peptides: Sequences shared between consecutive peptides are underlined. The PAR2 set also included a longer peptide centred around the consensus cleavage site. The consensus cleavage sites Arg41 and Arg36 are shown in white text on a black background. Peptides were incubated for 15 h with 150 nM each human granzyme and cleavage assessed by HPLC. PAR1 cleavage was observed on GzmB treatment of peptide 3. PAR2 cleavage was observed on GzmA and GzmK treatment of peptides 6 and 10. White arrowhead indicates uncleaved peptide; black arrowhead indicates cleaved peptide.

## Gzms A, B and K cleave PAR-based peptides

In conjunction with the NanoLuc® constructs, we also tested the granzymes' ability to cleave peptides corresponding to the N-terminal domains of PAR1 and PAR2. To accomplish this, we screened granzymes A, B and K against overlapping peptides covering the complete N-terminal domains of PAR1 or PAR2 from the N-terminus to the transmembrane domain. Peptide cleavage was determined by HPLC after 15 h of treatment with 100 nM granzyme and the cleavage sites were identified using mass spectrometry.

For the PAR1 analysis, three overlapping peptides were designed, with the canonical cleavage site Arg41 located in peptide 2. Unexpectedly, neither GzmA nor GzmK cleaved any of the PAR1 peptides, despite cleaving the full length NanoLuc® constructs. This suggests the requirement for further protein-protein interactions between PAR1 and an exosite on the granzyme surface to drive cleavage. Only GzmB cleaved a PAR1 peptide, and this cleavage occurred in peptide 3, which is significantly downstream of the canonical cleavage site (Fig 1C). This suggests that it represents a potential receptor disarming cleavage event.

For the PAR2 analysis we utilised six overlapping peptides from the initiator methionine to the beginning of the first transmembrane domain (Fig 1D). The PAR2 canonical P1 residue Arg36 was located in peptide 6, as well as in peptide 10, a longer sequence surrounding the canonical P1 site. Only the tryptic proteases, GzmA and GzmK, cleaved PAR2 peptides. Cleavage of peptide 6 and peptide 10 was observed (Fig 1D). LC-MS analysis indicated that cleavage by both proteases occurred at the canonical P1 residue Arg36.

As reactions were performed under identical conditions, a direct comparison of cleavage efficiencies could be made. For the shorter peptide 6, the cleaved peptide represented 1.6% of total for GzmA, but 9.6% of total for GzmK, while for the longer peptide 10 the cleavage product represented 6.6% of total for GzmA and 53% of total for GzmK. The increased activity on peptide 10 suggests an interaction of affinity-determining residues four or more residues downstream of the PAR cleavage site with an exosite on the protease. Taken with the NanoLuc®-PAR-eYFP cleavage data this indicates that GzmK is more efficient than GzmA at activating PAR2.

## GzmK induces biased PAR2 signalling

Given that cleavage of PAR1 by GzmK has been previously well described in endothelial cells [9], fibroblasts [10] and macrophages [8], we focused on the signalling mechanism and physiological consequences of GzmK cleavage of PAR2.

Cleavage of PAR2 at the canonical P1 site by trypsin leads to multiple intracellular signalling events such as $Ca^{2+}$ mobilisation, ERK phosphorylation and β-arrestin recruitment. To determine whether GzmK elicits similar signalling pathways to trypsin, we used epithelial KNRK cells producing very low levels of endogenous PAR and stably expressing transfected PAR2 [12].

Consistent with the literature, trypsin (10 nM) treatment of KNRK cells expressing PAR2 led to rapid and transient $Ca^{2+}$ mobilisation (Fig 2A and 2B). However, GzmK treatment did not result in a $Ca^{2+}$ flux, which was unexpected given that both the peptide and full-length

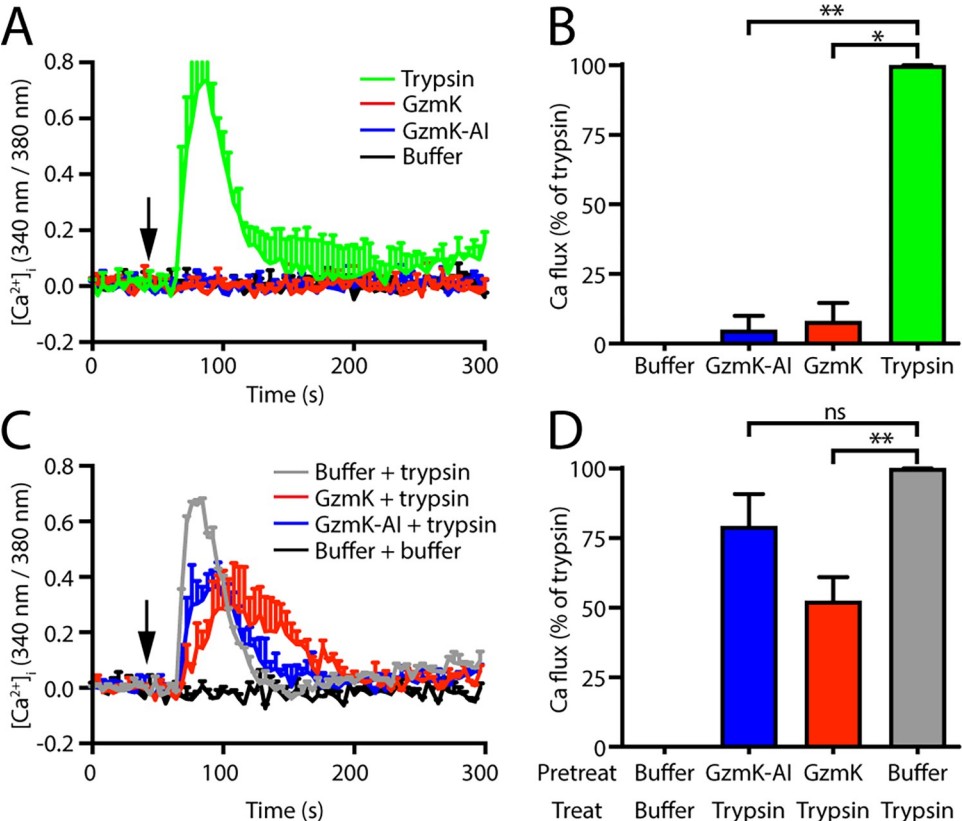

**Fig 2. Cleavage of PAR2 by granzyme K does not induce Ca²⁺ flux.** KNRK cells expressing human PAR2 (KNRK-PAR2) were treated with the indicated proteases. (A) Fluctuations in intracellular calcium were measured by change in Fura2 fluorescence spectra following treatment. Cells were treated with trypsin, granzyme K (GzmK) or AEBSF-inhibited GzmK (GzmK-AI) or buffer at the indicated time (arrow) and fluorescence measured over a 4 min window. (B) Total response, defined as the area under the curve was determined and standardised to the trypsin response. (C) Cells were pre-treated with GzmK, AEBSF-inhibited GzmK (GzmK-AI), or buffer for 30 min then treated with trypsin and the calcium flux followed by Fura2 fluorescence. (D) Total response, defined as the area under the curve was determined and standardised to the pre-treatment with vehicle response. *, p < 0.05; **, p < 0.01; ns, p > 0.05.

receptor cleavage data show that it cleaves PAR2 at the same site as trypsin (Fig 1). Although GzmK did not produce a PAR2-mediated Ca²⁺ flux, we considered the possibility that it may have the capacity to interfere with the trypsin-induced response. This was assessed in cells pre-treated with GzmK or buffer alone for 30 min prior to treatment with trypsin (Fig 2C and 2D). We observed a 50% decrease in trypsin-mediated Ca²⁺ release following GzmK treatment (Fig 2C and 2D). GzmK inhibited with the irreversible serine protease inhibitor AEBSF (GzmK-AI) did not affect trypsin signalling, indicating that the suppressing effect requires GzmK activity and is not due to steric interference.

In addition to Ca²⁺ signalling, activation of PAR2 by trypsin also leads to pERK activation, which involves β-arrestin-dependent receptor endocytosis [15]. GzmK treatment triggered both phosphorylation of ERK and recruitment of β-arrestin to a similar extent as trypsin (Fig 3). Activation of ERK in KNRK cells stably expressing PAR2 (KNRK-PAR2), but not the parental line (KNRK), confirmed that ERK signalling is PAR2-dependent (Fig 3A). GzmK activation of PAR2 mediated ERK phosphorylation to a similar extent as trypsin, whereas the inactivated GzmK did not show any ERK response (Fig 3A). Dose response experiments

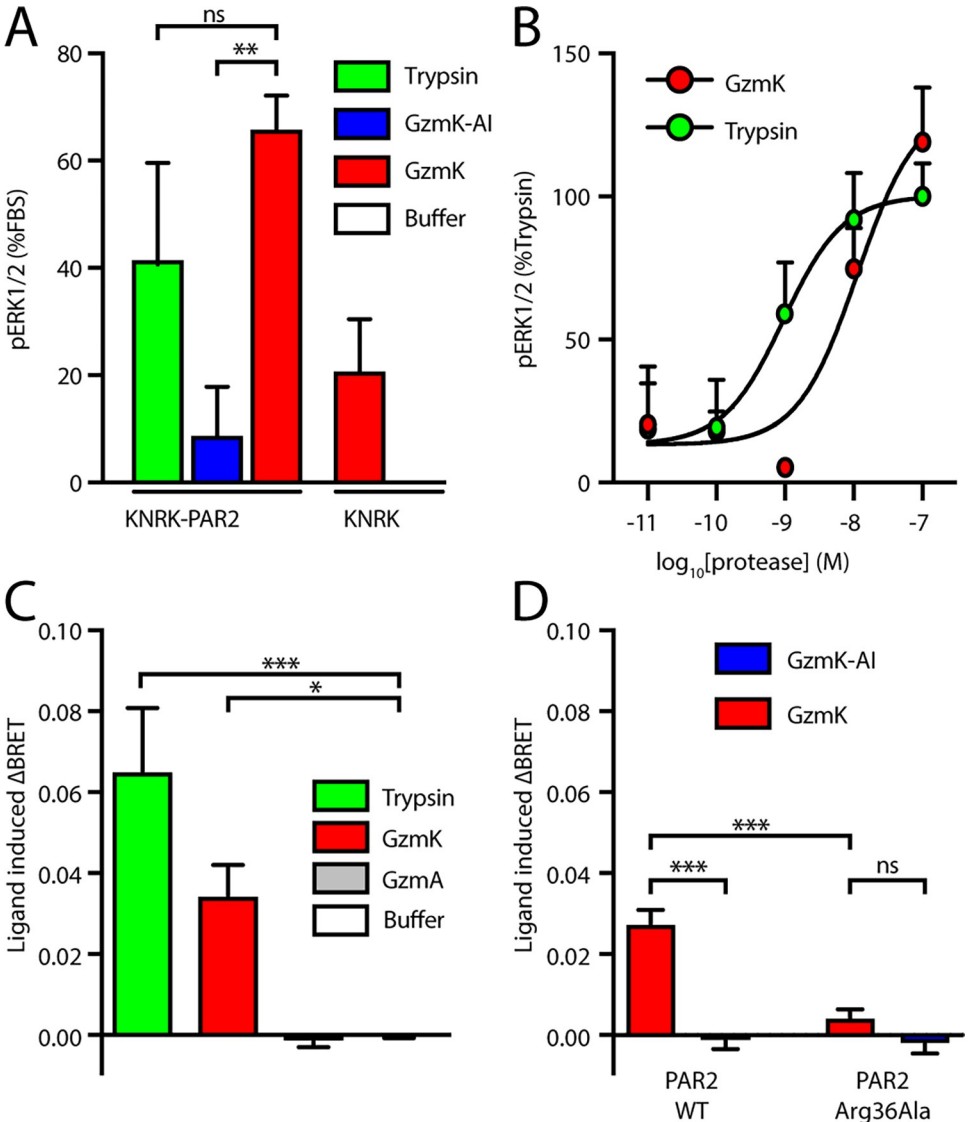

**Fig 3. Downstream signalling from PAR2 following cleavage by granzyme K.** (A) KNRK cells or KNRK-PAR2 cells were treated with the indicated proteases in serum-free medium and ERK phosphorylation was measured via Alpha-Screen. Data were normalised to the response to medium containing foetal calf serum. (B) ERK phosphorylation induced by a dilution series of granzyme K or trypsin. (C) KNRK cells transfected with PAR2-Rluc and β-arrestin-eYFP. Recruitment of β-arrestin to PAR2 by trypsin or GzmK was measured by BRET ratio. ***, $p < 0.001$; **, $p < 0.01$; *, $p < 0.05$; ns, $p > 0.05$.

showed that both GzmK and trypsin induce ERK phosphorylation with a potency ($EC_{50}$) in the low nanomolar range (Fig 3B).

Finally, we investigated the ability of GzmK to recruit arrestin to PAR2. To do this, the bioluminescence resonance energy transfer (BRET) signal between PAR2-Rluc8 and β-arrestin$_1$-YFP was monitored in HEK 293 cells. Our results show that GzmK (100 nM), but not GzmA, stimulates β-arrestin recruitment to a similar extent to trypsin (10 nM) (Fig 3C). The catalytic activity of GzmK is required for this recruitment as the inactive mutant GzmK failed to trigger β-arrestin recruitment (Fig 3C). Mutation of the P1 site Arg 36 severely diminished the recruitment of β-arrestin (Fig 3D), in agreement with earlier data using peptides and N-terminally tagged PAR2 constructs (Fig 1).

Overall, these data show that GzmK activates PAR2, initiating a biased signalling event that includes activation of ERK and recruitment of β-arrestin.

## GzmK-mediated interleukin release from epithelial cells involves PAR2

We next investigated the physiological effects of PAR activation by GzmK in an endogenous setting using epithelial A549 cells. We first used activating peptides corresponding to the tethered ligand of PAR1 or PAR2 to determine which PARs respond in our A549 cells. The PAR1 peptide (Thr-Phe-Leu-Leu-Arg-$NH_2$) induced a weak $Ca^{2+}$ response, whereas the PAR2 peptide (Ser-Leu-Ile-Gly-Lys-Val-$NH_2$) response was much stronger and sustained for longer (Fig 4). Given these peptides have documented similar potency at activating their respective receptors, our data suggest that PAR2 activity predominates in these cells.

To further explore the physiological effects of granzymes on A549 cells, we treated the A549 cells for 24 h with increasing amounts of GzmA or GzmK and assayed the culture medium for cytokine release. Both GzmA and GzmK induced the release of IL-8, while GzmK but not GzmA induced release of IL-6 (Fig 5A and 5B). Again, GzmK induced substantially (9-fold) more IL-8 release than GzmA, indicating that the cleavage efficiencies identified in the peptide cleavage and luciferase release assays are biologically relevant. In comparing relative levels of cytokine release, we found that GzmK induced release of approximately 9-fold more IL-8 than IL-6.

We next investigated the intracellular signalling events leading to IL-8 release using a series of small molecule inhibitors of PAR1 (vorapaxar), PAR2 (GB88), or MEK signalling (PD184352) (Fig 5C). The release of IL-8 in response to the PAR2-activating peptide was significantly inhibited by PD184352 and abolished by GB88. In cells treated with GzmK, IL-8 release was inhibited 70–80% by GB88 or PD184352, but was unaffected by vorapaxar (Fig 5C), indicating that release is mediated via PAR2 signalling through a MEK-dependent mechanism, and does not involve PAR1. Unlike PAR2 peptide treatment, GB88 did not completely abrogate the response induced by GzmK. This may be due to the higher stability of the protease compared to the PAR2 peptide (or GB88), allowing it to signal for longer.

## Discussion

We have shown here, using both synthetic peptides and full-length receptors, that granzymes A, B and K are able to cleave the activation domains of specific PARs. GzmA and GzmK cleave both PAR1 and PAR2, while GzmB cleaves PAR1 but was not tested on PAR2. GzmK is much more efficient at cleaving PAR1 and PAR2 than GzmA, and cleaves both receptors at the canonical site. Activation of PAR2 by GzmK does not trigger $Ca^{2+}$ mobilisation but initiates a biased signalling pathway that involves β-arrestin recruitment and ERK phosphorylation. In lung epithelial cells, GzmK treatment leads to the release of proinflammatory cytokines IL-6 and IL-8, the latter via a PAR2 and MEK-dependent mechanism.

GzmB cleavage of PAR1 was expected from literature reports [6,16]. The critical cleavage site-determining residues for GzmB are an Ile at P4 and Asp at P1 [17]. This is consistent with the motif Ile-Ser-Glu-Asp91 found in the most C-terminal PAR1 peptide tested here, well downstream of the canonical site at Arg41. Cleavage at Asp91 would almost certainly inactivate the receptor. This contradicts previous suggestions that cleavage of PAR1 by GzmB activates the receptor on neurons [6]. However, previous investigations of GzmB have indicated the presence of exosites that facilitate cleavage at suboptimal sites, particularly in extracellular matrix proteins [18–20]. As these still result in cleavage at acidic residues, the only potentially sub-optimal activating P1 residue in PAR1 would therefore be Asp39, two residues upstream of the canonical site. Asp39 is cleaved by matrix metalloprotease 1 to activate PAR1 on the

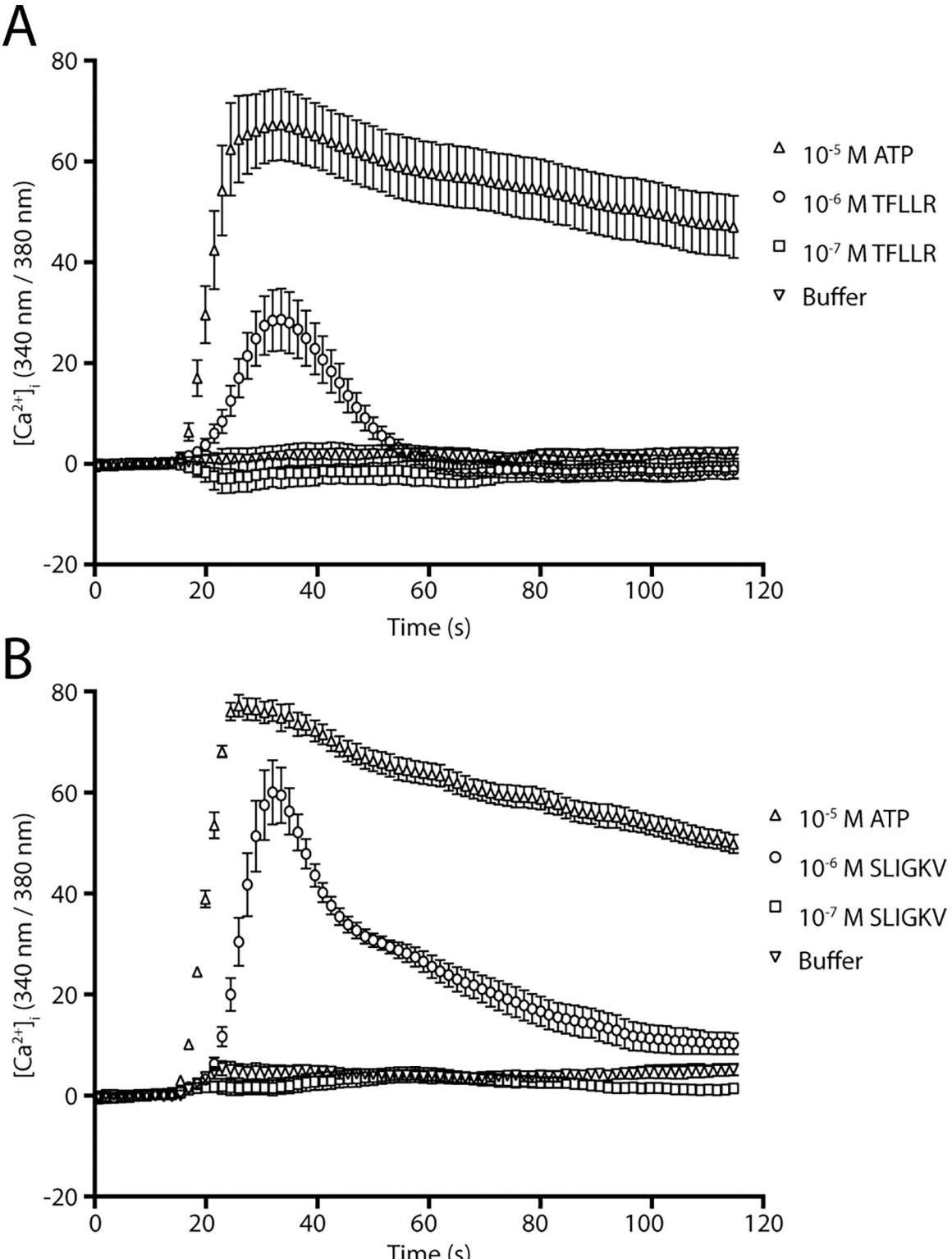

**Fig 4. Effect of PAR1 or PAR2 activating peptides on A549 cells.** A549 cells were loaded with Fura2 and treated with activating peptides for (A) PAR1 (TFLLR-NH$_2$), or (B) PAR2 (SLIGKV-NH$_2$) and calcium flux measured by the change in fluorescence and compared to ATP.

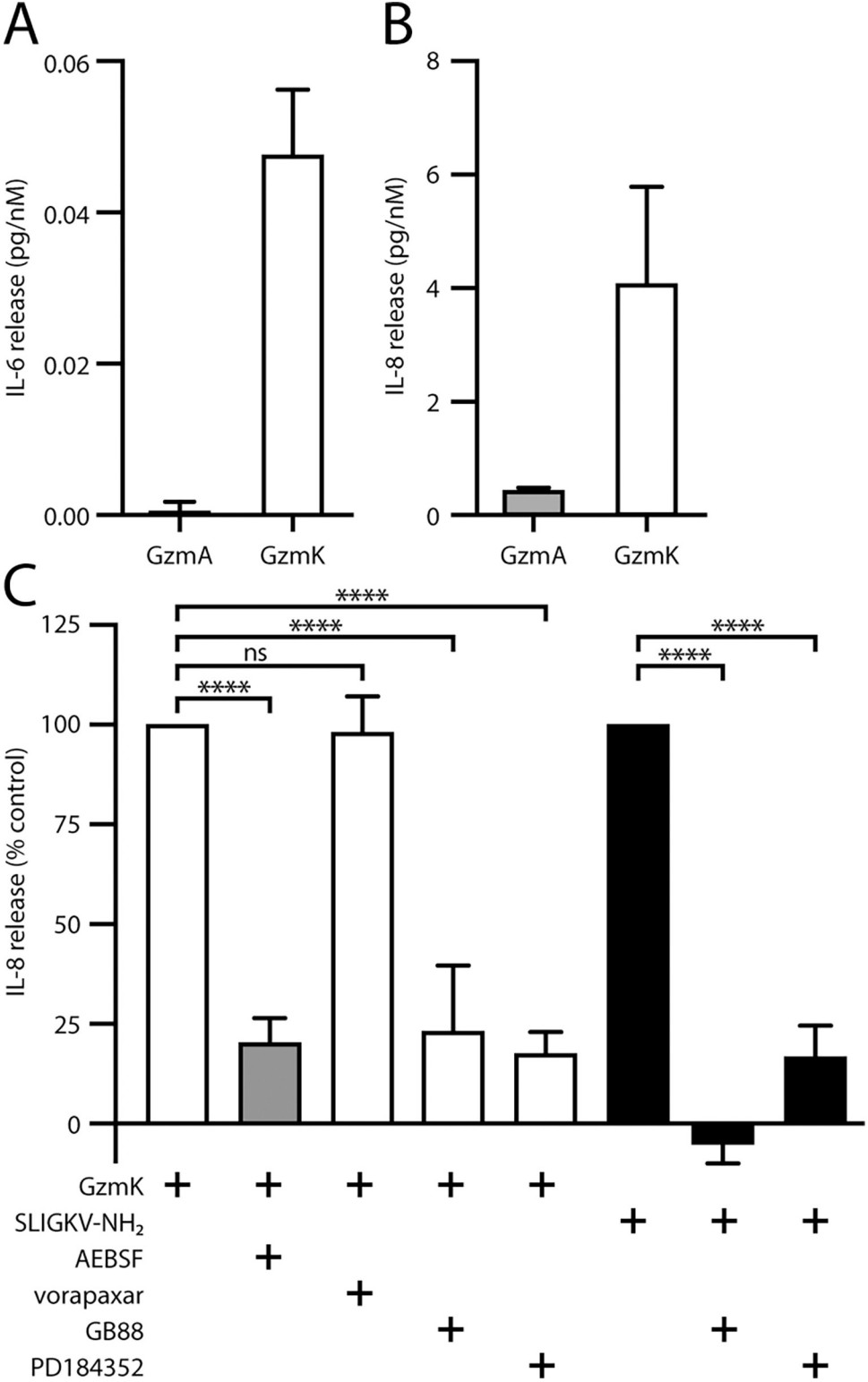

**Fig 5. Granzyme K induces chemokine release from A549 cells.** A549 cells were treated overnight with a serial dilution of either GzmA or GzmK in serum free medium and cell free supernatants were assayed for either (A) IL-6, or (B) IL-8 by ELISA. (C) A549 cells were pre-treated with inhibitors of PAR1 (vorapaxar), PAR2 (GB88) or MEK signalling (PD184352), then treated with either GzmK (white bars), GzmK-AI (grey bars) or PAR2 activating peptide (black bars) overnight in serum free medium. IL-8 release was measured by ELISA from cell-free supernatants. ****, $p < 0.0001$; ns, $p > 0.05$.

surface of platelets [21]. Therefore, it is possible that GzmB activity is directed to this site by exosite interactions with the full length PAR1, leading to receptor activation, in agreement with previous neuronal studies [6].

Despite previous speculation about redundant functions of GzmA and GzmK, the work here supports the suggestion that they have distinct biological roles. Both the peptide and full-length receptor cleavage data indicate the GzmK is much more efficient than GzmA at cleaving PAR2. This is possibly due to the presence of Leu at the P2' position, which has been shown to enhance cleavage by GzmK over GzmA [22]. Both GzmA and GzmK cleaved PAR2 peptides at the consensus site, but only cleaved PAR1 in the context of a complete receptor. This suggests a requirement for an exosite on the granzyme to participate in binding to PAR1. Indeed, exosite interactions are known to be an extremely strong component in selecting GzmA substrates in fully folded proteins [23], to the extent that the efficiency of cleavage of a particular substrate sequence differs if it is in a full-length protein compared to a peptide [24].

The downstream signalling effects of PAR cleavage are mediated by the tethered ligand of the neo-amino terminus. Curiously, GzmK and trypsin elicit differential signalling from PAR2 despite cleaving at the canonical P1 residue Arg36. While both proteases lead to ERK phosphorylation and β-arrestin recruitment, only trypsin initiates a $Ca^{2+}$ flux. A similar distinction was seen in the downstream signalling of PAR1 cleaved by GzmK, when compared to thrombin [9]. Biased PAR signalling has been observed with many proteases including elastase, proteinase 3, and various cathepsins [3]. Normally, the observed differences in signalling cascades require cleavage of a PAR at unique non-canonical sites, resulting in unique tethered ligands. This is the case for elastase activation of PAR1 and PAR2 [25–27], cathepsin S activation of PAR2 [13], and activated protein C (APC) activation of PAR1 [28]. How GzmK produces a different outcome while cleaving at the same site remains unclear at present. Although GzmK and trypsin cleave PAR2 at the same site on the receptor, GzmK may stabilise a distinct receptor conformation, which leads to the observed biased signalling profile. This could potentially be due to exosite binding between Gzms and the receptor.

It is interesting that, although GzmK failed to trigger PAR-2 mediated $Ca^{2+}$ mobilisation, pre-incubation of GzmK significantly reduced trypsin mediated $Ca^{2+}$ signalling. It is possible that by binding and cleaving PAR2, GzmK is able to trigger receptor internalisation (which is supported by its ability to recruit β-arrestin to the receptor); or that the binding between the exosite on the receptor and GzmK may prevent efficient cleavage by trypsin of PAR2. Which of these mechanisms contributes to the inhibitory effect of GzmK on trypsin signalling warrants further investigation.

GzmK triggers pERK1/2 activation in a concentration dependent manner, with similar potency to trypsin. pERK1/2 activation is an important downstream signalling endpoint upon PAR2 activation. pERK activation could be mediated via G protein-dependent and/or arrestin-dependent mechanisms [15]. Consistent with this, we have shown that GzmK induces β-arrestin recruitment to PAR2, suggesting a potential involvement of arrestin in GzmK-mediated pERK activation. However, we cannot rule out the participation of other mechanisms. For example, Rho kinase activation contributes to pERK activation during elastase-mediated PAR2 signalling [25]. Whether this pathway also contributes during GzmK-mediated PAR2 signalling remains unclear.

Finally, we have shown here that cleavage of PAR2 by GzmK results in the selective release of the cytokines IL-6 and IL-8 from epithelial cells. The Granville group has also shown that GzmK-mediated PAR1 cleavage induces the release of IL-6 and IL-8 from fibroblasts and endothelial cells to promote an inflammatory environment [8–10]. GzmK-induced PAR1 cleavage can also improve wound healing by keratinocytes [8]. Taken together, these observations raise the possibility that GzmK has redundant functions, or that it has evolved to elicit

similar, specific responses in multiple environments irrespective of the particular PAR expressed.

The biological significance of GzmK is unresolved. GzmK is not an efficient cytotoxin, as micromolar concentrations are required to induce death [29], whereas nanomolar concentrations of GzmB induce death [17]. Furthermore, mice lacking GzmK appear totally healthy, resist viral infection, and have no defect in killing *in vivo* or *in vitro* [30]. Our finding that GzmK via PAR2 selectively induces IL-6 and IL-8 release supports previous suggestions that it has a pro-inflammatory role [31,32]. Among other functions, IL-6 stimulates acute phase protein synthesis and neutrophil production, whereas IL-8 primarily acts as neutrophil attractant. Thus, GzmK may serve as a primary alarmin or an amplifier to support neutrophil recruitment to an injury site. Interestingly, GzmK is found in the circulation of healthy and diseased patients [33], and levels correlate with both disease severity and circulating cytokines during lung infection [34].

## Conclusion

Here we have shown that multiple granzymes have the ability to activate PAR1 and PAR2. We have extended the previous observations that GzmK activates PAR1, by showing that it also cleaves PAR2 on the surface of epithelial cells. This initiates signalling cascades involving recruitment of β-arrestin and phosphorylation of ERK, but not intracellular $Ca^{2+}$ fluxes. The physiological result of this signalling is the release of inflammatory cytokines IL-6 and IL-8 from cells. These results provide further support to the argument that GzmK acts as an inflammatory regulator *in vivo*.

## Supporting information

**S1 Data. Primary data.**
(XLSX)

## Author Contributions

**Conceptualization:** Morley D. Hollenberg, Nigel Bunnett, Andreas Suhrbier, Phillip Ian Bird.

**Data curation:** Dion Kaiserman, Nicholas Barlow.

**Formal analysis:** Dion Kaiserman, Peishen Zhao.

**Funding acquisition:** Andreas Suhrbier.

**Investigation:** Dion Kaiserman, Peishen Zhao, Caitlin Lorraine Rowe, Andrea Leong, Nicholas Barlow, Lars Thomas Joeckel, Phillip Ian Bird.

**Methodology:** Peishen Zhao, Nicholas Barlow, Lars Thomas Joeckel, Phillip Ian Bird.

**Project administration:** Dion Kaiserman.

**Resources:** Peishen Zhao, Nicholas Barlow, Corinne Hitchen, Sarah Elizabeth Stewart, Morley D. Hollenberg, Nigel Bunnett, Andreas Suhrbier, Phillip Ian Bird.

**Writing – original draft:** Dion Kaiserman, Peishen Zhao, Nicholas Barlow.

**Writing – review & editing:** Dion Kaiserman, Peishen Zhao, Caitlin Lorraine Rowe, Nicholas Barlow, Lars Thomas Joeckel, Corinne Hitchen, Sarah Elizabeth Stewart, Morley D. Hollenberg, Nigel Bunnett, Andreas Suhrbier, Phillip Ian Bird.

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
