## [Decision Letter · Decision Letter 0]

5 Jan 2022

PONE-D-21-39358Granzyme K initiates IL-6 and IL-8 release from epithelial cells by activating protease-activated receptor 2PLOS ONE

Dear Dr. Kaiserman,

Thank you for submitting your manuscript to PLOS ONE. After careful consideration, we feel that it has merit but does not fully meet PLOS ONE’s publication criteria as it currently stands. Therefore, we invite you to submit a revised version of the manuscript that addresses the points raised during the review process.

We look forward to receiving your revised manuscript.

Kind regards,

Karl X Chai, Ph.D.

Academic Editor

PLOS ONE

Journal Requirements:

"This work was supported by the National Health and Medical Research Council Australia (Project Grant 1141421 and Program Grant 490900), National Institutes of Health (NS102722, DE026806, DK118971, DE029951) and Department of Defense (W81XWH1810431)."

"This work was funded by a National Health and Medical Research Council Australia Project Grant 1141421 (PIB & AS), a National Health and Medical Research Council Australia Program Grant 490900 (PIB), National Institutes of Health grants NS102722, DE026806, DK118971 and DE029951 (NB) and Department of Defense grant W81XWH1810431 (NB). The funders had no role in study design, data collection and analysis, decision to publish, or preparation of the manuscript."

6. Please amend the manuscript submission data (via Edit Submission) to include author Andrea Leong.

7. We note that you have included the phrase “data not shown” in your manuscript. Unfortunately, this does not meet our data sharing requirements. PLOS does not permit references to inaccessible data. We require that authors provide all relevant data within the paper, Supporting Information files, or in an acceptable, public repository. Please add a citation to support this phrase or upload the data that corresponds with these findings to a stable repository (such as Figshare or Dryad) and provide and URLs, DOIs, or accession numbers that may be used to access these data. Or, if the data are not a core part of the research being presented in your study, we ask that you remove the phrase that refers to these data.

Reviewers' comments:

Reviewer's Responses to Questions

**Comments to the Author**

1. Is the manuscript technically sound, and do the data support the conclusions?

Reviewer #1: Partly

2. Has the statistical analysis been performed appropriately and rigorously? 

Reviewer #1: Yes

3. Have the authors made all data underlying the findings in their manuscript fully available?

Reviewer #1: Yes

4. Is the manuscript presented in an intelligible fashion and written in standard English?

Reviewer #1: Yes

5. Review Comments to the Author

Reviewer #1: The manuscript by Kaiserman et al. “Granzyme K Activates PAR2” assesses a role for GzmK in PAR2 activation. Previous studies suggest a role for GzmK in PAR1 activation and proinflammatory cytokine release. The present study proposes a novel role for GzmK in PAR2 activation. Interestingly, although cleavage occurs at the canonical site similar to other trypsin, Ca2+ mobilisation is not initiated but rather induces ERK phosphorylation and b-

arrestin recruitment in epithelial cells. Historically Gzms have been thought to function exclusively as perforin-dependent, cytotoxic proteases.The present manuscript supports emerging evidence for a non-cytotoxic, extracellular roles for granzymes. A number of suggestions are listed below for improvement:

Major:

1. Introduction: Some key citations are missing eg. p4: “GzmK cleaves and activates PAR1 on endothelial cells and fibroblasts (Sharma et al., 2016; Turner et al., 2019)” should be amended to include Cooper et al., 2011. This should also be included in first line of Results “Previous studies have indicated that GzmA and GzmK can cleave and activate PARs (Suidan et al., 1994; Sower et al., 1996b; Sharma et al., 2016) given its relevance to the present article. Recent articles in Immunity, 2021 suggesting a role for GzmK in inflammaging as well as the recent paper citing a role for extracellular GzmB in maternally-transferred childhood asthma citing a mechanism involving PAR2 published in J Clin Invest (Qian et al, 2020) should also be mentioned possibly in the intro and/or discussion.

2. It is unclear as to why GzmB-mediated PAR2 activation was not explored given the high impact study published last year in the Journal of Clinical Investigation by Qian et al suggesting a key role for extracellular GzmB-mediated PAR2 activation of airway epithelial cells as the mechanism as to how maternal exposure to diesel exhaust particles predispose offspring to allergic airway disease.

3. Results are well-described although it is unclear as to why GzmB was dropped after PAR1. GzmB was also shown to induce IL-8 release via an unknown mechanism (Hiroyasu et al, Nature Comm, 2021). Given that GzmB was not assessed throughout the manuscript, it may be worth focussing exclusively on GzmA and K and dropping the GzmB data as it is incomplete. Alternatively, adding GzmB throughout for PAR2/IL6/8 studies could be added to give a more complete picture. The latter would require substantial revision.

4. Discussion Line 3 states “GzmA and GzmK cleave both PAR1 and PAR2, while GzmB cleaves only PAR1.” This contradicts the study by Qian et al (J Clin Invest, 2020) in which a substantial amount of evidence would suggest otherwise in a clinically-relevant model. This must be supported by data and a strong discussion if disputing this work given the impact. I would suggest removing the GzmB PAR2 work and any text dismissing GzmB-mediated PAR2 activation as this was not adequately assessed in the present manuscript.

Minor

1. Methods: Please include number of replicates where relevant.

2. Results: ‘Cleavage of full-length PAR1 and PAR2 by GzmA and GzmK’: Literature reports a sharp decrease in activity of the NanoLuc system below pH 8, whereas Gzms optimally function around pH 7. A brief mention of the potential limitations should be discussed.

3. Discussion: “In lung epithelial cells, GzmK treatment leads to the release of proinflammatory cytokines IL-6 and IL-8, but not TNFα or IL-1β, via a PAR2 and MEK-dependent mechanism.” Include reference.

4. Discussion: “GzmK treatment leads to the release of proinflammatory cytokines IL-6 and IL-8, but not TNFα or IL-1β, …” is inconsistent with the previous results section which mentioned GzmK treatment did not lead to IFN-y or IL-1β release (TNFα not mentioned). This sentence also mentioned that the mechanisms of IL-6 and IL-8 release were both PAR2 and MEK-dependent, however, the previous results section (and corresponding figure) only investigated this mechanism in the context of IL-8. It should be mentioned that GzmB can induce IL-8 release from keratinocytes (Hiroyasu et al, Nature Comm., 2021)

5. Figures, while one can interpret the results, are of low resolution/blurry in the PDF version.

6. Figures, 1a/b: Describe source of WT PARS. I may have missed where it was discussed in the text.

7. Figures, 2c/d: The experiment group trypsin + buffer is coloured grey instead of green as depicted in the legend.

8. Figures, 2-5: There are no statistics/p values included in the graphs despite significance values having been mentioned within the figure legends.

9. Figures 3: Since the same experimental groups are presented as in Figure 2, it would be ideal to keep colours/patterns consistent for treatments.

10. Figure 4: The figure legend is inconsistent with the presented images – there is no (c) or (d). As a result, the labels are also incorrect. There was no display of IL-8 release shown.

11. Figure 5: For clarity for the reader, include in brackets in the figure legend the identity of each PAR inhibitor, as done in text. Also, there was no mention in the figure legend of the MEKi (PD184352) or what the solid bars represent (treatment with GzmK vs activating peptide). Previous figures used the nomenclature GzmK-AI instead of AEBSF. For consistency, it should be the same.

6. PLOS authors have the option to publish the peer review history of their article (what does this mean?). If published, this will include your full peer review and any attached files.

Reviewer #1: No

---

## [Author Response · Author response to Decision Letter 0]

29 May 2022

Reviewers Comments

Major

1. The Cooper et al and Qian et al references have been included. The inflammaging paper (Immunity 2021) was already referenced in the discussion (Ref 32).

2. Granzyme B experiments were performed prior to the publication of the Qian et al paper and we are not in a position to perform them any more. As such, we have included references to the Qian paper and removed references to GzmB not cleaving PAR2.

3. The primary focus of this paper was GzmA and GzmK. GzmB was used in the luciferase assays as a control for cleavage away from the consensus site. We have made this more clear in the text. 

4. As described above, we have altered our references to GzmB and more clearly defined its role as a control in the earlier experiments.

Minor

1. This has been included

2. We disagree that the pH sensitivity of the reagents affects our analyses. All assays were performed in buffered medium at an optimal granzyme cleavage pH (7.4). As strong responses were observed for GzmK, the poor responses for GzmA and GzmB are due to poor cleavage of the protein substrate rather than pH incompatibilities of the system.

3. Data deleted.

4. This lead in paragraph is a brief summary of our results. The wording has been improved.

5. High resolution images for publication included.

6. Described in the methods as coming from Prof Hollenberg (U Calgary)

7. The figure legend was incorrect. It has been fixed.

8. P values included on graphs.

9. This has been done.

10. This has been fixed.

11. This has been fixed.

Editors Comments

1. The text file has been reformatted.

2. The reference list is correct.

3. The funding information has been removed from the manuscript. I cannot find our original funding information, however I can confirm that the funding statement is correct.

"This work was funded by a National Health and Medical Research Council Australia Project Grant 1141421 (PIB & AS), a National Health and Medical Research Council Australia Program Grant 490900 (PIB), National Institutes of Health grants NS102722, DE026806, DK118971 and DE029951 (NB) and Department of Defense grant W81XWH1810431 (NB). The funders had no role in study design, data collection and analysis, decision to publish, or preparation of the manuscript."

4. The underlying primary data has been provided as a supplementary file.

5. Andrea has been added.

6. The reference to data not shown has been removed.

---

## [Decision Letter · Decision Letter 1]

14 Jun 2022

Granzyme K initiates IL-6 and IL-8 release from epithelial cells by activating protease-activated receptor 2

PONE-D-21-39358R1

Dear Dr. Kaiserman,

We’re pleased to inform you that your manuscript has been judged scientifically suitable for publication and will be formally accepted for publication once it meets all outstanding technical requirements.

Kind regards,

Karl X Chai, Ph.D.

Academic Editor

PLOS ONE

Additional Editor Comments (optional):

Reviewers' comments:

Reviewer's Responses to Questions

**Comments to the Author**

1. If the authors have adequately addressed your comments raised in a previous round of review and you feel that this manuscript is now acceptable for publication, you may indicate that here to bypass the “Comments to the Author” section, enter your conflict of interest statement in the “Confidential to Editor” section, and submit your "Accept" recommendation.

Reviewer #1: All comments have been addressed

2. Is the manuscript technically sound, and do the data support the conclusions?

Reviewer #1: Yes

3. Has the statistical analysis been performed appropriately and rigorously? 

Reviewer #1: I Don't Know

4. Have the authors made all data underlying the findings in their manuscript fully available?

Reviewer #1: Yes

5. Is the manuscript presented in an intelligible fashion and written in standard English?

Reviewer #1: Yes

6. Review Comments to the Author

Reviewer #1: The authors have adequately addressed all of my concerns. The research offers further insights into the potential mechanism of action for granzyme K in PAR-2 activation.

7. PLOS authors have the option to publish the peer review history of their article (what does this mean?). If published, this will include your full peer review and any attached files.

Reviewer #1: No

---

## [Editor Report · Acceptance letter]

18 Jul 2022

PONE-D-21-39358R1 

Granzyme K initiates IL-6 and IL-8 release from epithelial cells by activating protease-activated receptor 2 

Dear Dr. Kaiserman:

I'm pleased to inform you that your manuscript has been deemed suitable for publication in PLOS ONE. Congratulations! Your manuscript is now with our production department. 

Kind regards, 

on behalf of

Dr. Karl X Chai 

Academic Editor

PLOS ONE